# The impact of longstanding illness and common mental disorder on competing employment exits routes in older working age: A longitudinal data-linkage study in Sweden

Lisa Harber-Aschan[1]*, Wen-Hao Chen[2], Ashley McAllister[1,3], Natasja Koitzsch Jensen[4], Karsten Thielen[4], Ingelise Andersen[4], Finn Diderichsen[4], Ben Barr[5], Bo Burström[1]

1 Department of Global Public Health, Equity and Health Policy Research Group, Karolinska Institutet, Stockholm, Sweden, 2 Social Analysis and Modelling Division, Statistics Canada, Ottawa, Canada, 3 Melbourne School of Global and Population Health, Disability and Health Unit, The University of Melbourne, Melbourne, Australia, 4 Department of Public Health, University of Copenhagen, Copenhagen, Denmark, 5 Department of Public Health and Policy, University of Liverpool, Liverpool, England, United Kingdom

* lisa.harber-aschan@ki.se

**Data Availability Statement:** The data used in the analyses of this article are coded individual survey

## Abstract

### Objectives

Comorbidity is prevalent in older working ages and might affect employment exits. This study aimed to 1) assess the associations between comorbidity and different employment exit routes, and 2) examine such associations by gender.

### Methods

We used data from employed adults aged 50–62 in the Stockholm Public Health Survey 2002 and 2006, linked to longitudinal administrative income records (N = 10,416). The morbidity measure combined Limiting Longstanding Illness and Common Mental Disorder—captured by the General Health Questionnaire-12 ($\geq$4)—into a categorical variable: 1) No Limiting Longstanding Illness, no Common Mental Disorder, 2) Limiting Longstanding Illness only, 3) Common Mental Disorder only, and 4) comorbid Limiting Longstanding Illness +Common Mental Disorder. Employment status was followed up until 2010, treating early retirement, disability pension and unemployment as employment exits. Competing risk regression analysed the associations between morbidity and employment exit routes, stratifying by gender.

### Results

Compared to No Limiting Longstanding Illness, no Common Mental Disorder, comorbid Limiting Longstanding Illness+Common Mental Disorder was associated with early retirement in men (subdistribution hazard ratio = 1.73, 95% confidence intervals: 1.08–2.76), but not in

participant data with linked data from national registries. These data are available to the extent permitted by national and EU legislation upon request from Karolinska Institutet's Research Data Office (rdo@ki.se) from the date of publication until 10 years after the date of publication.

**Funding:** The study is part of a larger project entitled Tackling Health Inequalities and Extending Working Lives (THRIVE), within the framework of the Joint Programming Initiative More Years Better Lives. This work was supported by The Innovation Fund Denmark (https://eur01.safelinks.protection. outlook.com/?url=https%3A%2F% 2Finnovationsfonden.dk%2Fen&data=02%7C01% 7Clisa.harber-aschan%40ki.se% 7Cc0719da96d314803341a08d7a99b2389% 7Cbff7eef1cf4b4f32be3da1dda043c05d%7C0% 7C0%7C637164358617794237&sdata=JycvAft2w OpBKlvmjP6ZLOerqswabq0DGNkH1IhEn9w% 3D&reserved=0; 5194-00004B, awarded to FD and IA), the Swedish Research Council for Health, Working Life and Welfare (https://eur01.safelinks. protection.outlook.com/?url=https%3A%2F% 2Fforte.se%2Fen%2F&data=02%7C01%7Clisa. harber-aschan%40ki.se% 7Cc0719da96d314803341a08d7a99b2389% 7Cbff7eef1cf4b4f32be3da1dda043c05d%7C0% 7C0%7C637164358617794237&sdata= kguLS3jcMh9Tk%2BUalWoXYDjn2WVvEfpkJY5% 2FwMDYzEQ%3D&reserved=0; 2015-01531 awarded to Bo Burström and AM), The UK Economic and Social Research Council (https:// eur01.safelinks.protection.outlook.com/?url=https %3A%2F%2Fesrc.ukri.org%2F&data=02%7C01% 7Clisa.harber-aschan%40ki.se% 7Cc0719da96d314803341a08d7a99b2389% 7Cbff7eef1cf4b4f32be3da1dda043c05d%7C0% 7C0%7C637164358617794237&sdata= gqKtq0uvbl6imJJ5v%2FM2w4T0DbfSj5% 2FfJMDCeafk7s4%3D&reserved=0; ES/N019261/1 awarded to Ben Barr), The Canadian Institutes of Health Research (https://eur01.safelinks. protection.outlook.com/?url=http%3A%2F% 2Fwww.cihr-irsc.gc.ca%2Fe%2F193.html&data= 02%7C01%7Clisa.harber-aschan%40ki.se% 7Cc0719da96d314803341a08d7a99b2389% 7Cbff7eef1cf4b4f32be3da1dda043c05d%7C0% 7C0%7C637164358617794237&sdata= YsnCGtuOCCLWk%2Fnk3l%2Fw3xsMrkIf3Y4qEnr %2BKeKD3tE%3D&reserved=0; 2016-18). The funders had no role in study design, data collection and analysis, decision to publish, or preparation of the manuscript.

**Competing interests:** The authors have declared that no competing interests exist.

women. For men and women, strong associations for disability pension were observed with Limiting Longstanding Illness only (subdistribution hazard ratio = 11.43, 95% confidence intervals: 9.40–13.89) and Limiting Longstanding Illness+Common Mental Disorder (subdistribution hazard ratio = 14.25, 95% confidence intervals: 10.91–18.61), and to a lesser extent Common Mental Disorder only (subdistribution hazard ratio = 2.00, 95% confidence intervals: 1.31–3.05). Women were more likely to exit through disability pension than men (subdistribution hazard ratio = 1.96, 95% confidence intervals: 1.60–2.39). Common Mental Disorder only was the only morbidity category associated with unemployment (subdistribution hazard ratio = 1.70, 95% confidence intervals: 1.36–2.15).

## Conclusions

Strong associations were observed between specific morbidity categories with different employment exit routes, which differed by gender. Initiatives to extend working lives should consider older workers' varied health needs to prevent inequalities in older age.

## Introduction

Many countries are introducing extending working lives policies as means of addressing the increasingly ageing population, which presents challenges to maintaining welfare systems [1]. Poor health is a major determinant of early labour market exit, and unpacking this association is necessary to inform policies for extending working lives [2]. Longstanding illnesses, such as cardiovascular disease, diabetes and musculoskeletal conditions, are strongly associated with premature labour markets exits [3–5]. Depression and anxiety, referred to as common mental disorders (CMD), are also recognised as important determinants of non-employment, and have recently been found to be specifically associated with unemployment in older workers [6–8]. The comorbidity of longstanding illnesses and CMD has not been extensively studied, but recent evidence suggests that the combination of mental and physical illness may make workers vulnerable to employment exits such as early retirement and disability pension [9,10]. This issue is particularly relevant among older workers since comorbidity is more prevalent in this group [11], and may increase the likelihood of employment exits due to greater disability and poorer occupational functioning [12,13]. Studies examining comorbidity as a broad construct suggest that it affects older workers' employment exits, possibly in a dose-response manner [6,8,14–17], and additive effects of depression and heart disease on labour market participation have also been observed [18]. Yet, others have found no effect of depressive symptoms among those with chronic disease on working until retirement [4]. Evidence suggest that various indicators of health have differential effects on early retirement, disability pension and unemployment, but comorbidity is not well understood in relation to specific employment exit pathways among older workers [3,8,9].

Furthermore, older workers represent a heterogenous group, and the determinants of specific employment exit routes for different demographic groups remains under-explored, not least by gender [1]. Gender differences are important to examine given that men's and women's career trajectories are different over the life course, and women's retirement decisions are more likely to be influenced by household factors [19]. Moreover, morbidity patterns vary by gender and may differentially influence employment exit routes, but have not been extensively studied.

This study applied a longitudinal approach using a unique survey-administrative linked dataset for Stockholm, Sweden to examine the impact of comorbidity on employment exits. The study aimed to: 1) assess the effects of comorbid Limiting Longstanding Illness (LLI) and CMD on different employment exits routes, and 2) examine whether such associations varied by gender.

## Materials and methods

### Data

The study used a survey-administrative dataset from Sweden, linking the Stockholm County Council Public Health Survey for the years 2002 and 2006 to Swedish Populations Registers and the longitudinal integration database for health insurance and labour market studies (LISA) [20]. The survey randomly sampled residents in Stockholm County aged 18–84 and collected self-reported information about health, sociodemographic and social characteristics (response rates >60%) [21]. Compared to Stockholm census data, survey participants were more likely to be female, born in Sweden and have higher socio-economic status [21]. Survey data for consenting survey respondents were linked to aforementioned administrative data-bases, using the unique personal identification number for Swedish residents.

### Study design

The study applied a longitudinal open cohort study design. Cross-sectional survey data from 2002 and 2006 were baseline timepoints, and annual follow-up of employment exits were determined by administrative income records available until 2010. The linked survey-administrative data thus produced a person-years dataset with multiple follow-up points for each survey respondent. S1 Fig illustrates the process of deriving the analytical sample of N = 10,416 survey respondents. The sample selection criteria were: 1) aged 50–62 at baseline, 2) employed at baseline, and 3) at least 2 years of follow-up income records after baseline. We censored individuals at 65, which is a typical, although not obligatory, retirement age. We further restricted the sample to respondents with complete health and employment data. The study was approved by the Regional Ethical Review Board in Stockholm (Regionala Etikprövnings-nämnden (EPN) i Stockholm; 2016/1353-31/5). The study was a secondary data analysis which analysed data anonymously.

### Measures

**Morbidity.** CMD was captured using the 12-item General Health Questionnaire (GHQ-12), a screen assessing symptoms of psychological distress [22]. Informed by recent research validating the screen against psychiatric outpatient records, the ≥4 cut-off indicated CMD [23]. LLI was captured by a self-reported variable indicating whether the respondent had long-standing health problems that limit the ability to work or perform other daily activities. These binary measures produced a categorical exposure variable: 1) *No LLI or CMD* (GHQ<4; No LLI), 2) *LLI only* (GHQ<4; LLI), 3) *CMD only* (GHQ≥4; No LLI), and 4) *comorbid LLI+CMD* (GHQ≥4; LLI).

**Employment exit.** We defined employment as annual earnings of ≥60 000 SEK, from paid and self-employed income sources [24]. This figure approximately corresponds to 5700 EURO, or two months' full-time income. Employment records were available until 2010, producing a maximum of 8 and 4 follow-up points for the 2002 and 2006 samples, respectively. Employment exits distinguished between early retirement, disability pension and unemployment, using annual information from the LISA database [20]. Unemployment included those

who were registered with the national unemployment agency or receiving unemployment benefits. Disability pension was measured by disability pension benefits payments. Early retirement was defined as any retirement pension (either from state pension available from the age of 61, or from employer or private pension schemes available earlier), in combination with annual earnings of less than 60 000 SEK. If unemployment or early retirement was captured in the same year as disability pension, the employment exit was classified as disability pension; if unemployment and early retirement occurred in the same year, the employment exit was classified as early retirement.

**Covariates.** All covariates were captured at baseline. Measures obtained from the Swedish Population Registers included age, sex, education, country of birth, and marital status. Education distinguished between primary, secondary and university education. Country of birth grouped those born in Sweden and elsewhere. Marital status grouped those who were married or in registered partnership vs. not married (single/divorced/widowed).

The covariates obtained from the survey included self-rated health, social occupational class, financial strain, and employment conditions. Social occupational class recoded the 10 categories of the Swedish Standard Classification of Occupations 2012 [25] into four categories: 1: High non-manual, 2: Intermediate non-manual, 3: Low non-manual, and 4: Manual. Financial strain grouped those who had borrowed money from family/friends to afford food or rent in the past year, and those who had not. Employment conditions distinguished between those who were: 1: Employed with great freedom, 2: Employed with limited freedom, and 3: Self-employed. Freedom at work was assessed by two questions regarding freedom to decide: 1) what to do, and 2) how to perform their tasks on a 4-point scale (always/often/rarely/never). Those who indicated never or rarely to both items were classed as having "limited" freedom. All covariates had less than 1% missing data.

## Analysis

Descriptive statistics presented percentage estimates of the sample by morbidity status. Fine and Gray subdistribution hazard models estimated associations between morbidity categories and covariates and employment exits on the subdistribution hazard function, considering other competing exit routes [26]. Unadjusted and adjusted subdistribution hazard ratios (SHR) were estimated, indicating the relative change in the subdistribution hazard function according to morbidity or covariates. The direction of the SHR may be interpreted as a morbidity category's effect on the "incidence" of employment exit [27]. The *stcrreg* command was used in Stata 15 statistical software, applying complete case analysis [28]. Separate models were estimated for men and women. The results were presented visually by estimating the cumulative incidence functions (CIFs) by morbidity status, separately for men and women. CIFs estimated the probability of the employment exits of interest, in the presence of the two competing employment exits. Observations were censored at age 65.

## Results

Approximately 25% of older workers reported some illness: 15.3% had LLI only, 6.4% had CMD only, and 3.4% reported comorbid LLI+CMD; most had neither of these conditions (Table 1). Women were more likely to experience CMD only and comorbid LLI+CMD. Older respondents had more LLI only, and younger respondents reported more CMD only and LLI +CMD. LLI+CMD was also more prevalent among those born outside Sweden, non-married, having financial strain, and reporting limited perceived work freedom. Those with higher education and non-manual social occupational class reported more CMD only, while LLI only was more common among those with lower education and manual occupations. LLI+CMD

**Table 1. Descriptive statistics of full sample, and by morbidity status.**

| | Full sample (n, %) | | Morbidity status (n, row %) | | | | | | | |
|---|---|---|---|---|---|---|---|---|---|---|
| | (N = 10,416) | | No LLI, no CMD (n = 7790) | | LLI only (n = 1593) | | CMD only (n = 662) | | LLI+CMD (n = 358) | |
| **Sex** | | | | | | | | | | |
| Men | 4896 | 47.0 | 3796 | 77.6 | 751 | 15.4 | 230 | 4.7 | 114 | 2.3 |
| Women | 5520 | 53.0 | 3994 | 72.5 | 842 | 15.3 | 432 | 7.8 | 244 | 4.4 |
| **Age** | | | | | | | | | | |
| 50–54 | 4074 | 39.1 | 3065 | 75.3 | 550 | 13.5 | 305 | 7.5 | 150 | 3.7 |
| 55–59 | 4269 | 41.0 | 3147 | 73.9 | 693 | 16.3 | 266 | 6.2 | 155 | 3.6 |
| 60–62 | 2073 | 19.9 | 1578 | 76.2 | 350 | 16.9 | 91 | 4.4 | 53 | 2.6 |
| **Country born** | | | | | | | | | | |
| Sweden | 8945 | 85.9 | 6747 | 75.5 | 1357 | 15.1 | 562 | 6.3 | 284 | 3.2 |
| Outside Sweden | 1471 | 14.1 | 1043 | 71.0 | 250 | 16.8 | 105 | 7.1 | 76 | 5.2 |
| **Marital status** | | | | | | | | | | |
| Married | 6527 | 62.7 | 4994 | 76.3 | 967 | 14.7 | 390 | 5.9 | 203 | 3.1 |
| Single, divorced, widowed | 3889 | 37.3 | 2835 | 72.6 | 640 | 16.3 | 277 | 7.1 | 157 | 4.0 |
| **Education** | | | | | | | | | | |
| Primary | 1472 | 14.1 | 1072 | 73.0 | 283 | 19.3 | 65 | 4.4 | 49 | 3.3 |
| Secondary | 4380 | 42.1 | 3236 | 74.0 | 747 | 17.1 | 247 | 5.6 | 146 | 3.3 |
| University | 4554 | 43.8 | 3476 | 76.4 | 560 | 12.3 | 349 | 7.7 | 163 | 3.6 |
| **Social occupational class** | | | | | | | | | | |
| High non-manual | 3767 | 37.6 | 2904 | 77.2 | 451 | 12.0 | 283 | 7.5 | 123 | 3.3 |
| Intermediate non-manual | 2484 | 24.8 | 1838 | 74.1 | 375 | 15.1 | 173 | 7.0 | 95 | 3.8 |
| Low non-manual | 2252 | 22.5 | 1648 | 73.3 | 385 | 17.1 | 132 | 5.9 | 84 | 3.7 |
| Manual | 1523 | 15.2 | 1117 | 73.4 | 315 | 20.7 | 47 | 3.1 | 43 | 2.8 |
| **Employment conditions** | | | | | | | | | | |
| Employed with great work freedom | 8027 | 77.9 | 6191 | 76.0 | 1180 | 14.7 | 505 | 6.3 | 242 | 3.1 |
| Employed with limited work freedom | 1139 | 11.0 | 749 | 65.9 | 212 | 18.7 | 101 | 8.9 | 75 | 6.6 |
| Self-employed | 1144 | 11.1 | 873 | 76.4 | 186 | 16.3 | 48 | 4.2 | 35 | 3.1 |
| **Financial strain** | | | | | | | | | | |
| No | 9792 | 94.4 | 7415 | 75.8 | 1459 | 14.9 | 604 | 6.2 | 302 | 3.1 |
| Yes | 585 | 5.6 | 347 | 59.4 | 129 | 22.1 | 54 | 9.3 | 54 | 9.3 |

LLI; limiting longstanding illness, CMD: common mental disorder

was more prevalent among those of intermediate and low non-manual social class. The full analytical sample consisted of 10 416 respondents at baseline, and 68 642 person-year observations. Early retirement was the most common employment exit route, followed by unemployment, and disability pension (17.02, 12.44, and 8.92 per 1000 person-years, respectively).

## Early retirement

Overall, analysis of the full sample did not show any associations between neither LLI only, CMD only, nor LLI+CMD with early retirement (Table 2). However, stratification by gender indicated that comorbidity was associated with an increased subdistribution hazard rate of early retirement for men (SHR: 1.73), but not for women (SHR: 0.81) (Fig 1). Migrant men (SHR: 0.64) and single, divorced and widowed women (SHR: 0.58) had a lower subdistribution

**Table 2.** Adjusted competing risks analyses on the influence of health, demographic, and socio-economic factors and work conditions at baseline among employed persons on the likelihood of early retirement in the full sample, and stratified by gender.

| | Early retirement: Full sample (n = 1093/9811) | | | Early retirement: Men (n = 520/4600) | | | Early retirement: Women (n = 573/5211) | | |
|---|---|---|---|---|---|---|---|---|---|
| | SHR | (95% CI) | p | SHR | (95% CI) | p | SHR | (95% CI) | p |
| **Morbidity** | | | | | | | | | |
| No LLI, No CMD | 1.00 | | | 1.00 | | | 1.00 | | |
| LLI only | 0.90 | (0.76–1.07) | 0.223 | 1.00 | (0.79–1.27) | 0.975 | 0.83 | (0.66–1.05) | 0.126 |
| CMD only | 1.06 | (0.83–1.36) | 0.647 | 1.10 | (0.73–1.65) | 0.646 | 1.04 | (0.76–1.43) | 0.802 |
| LLI+CMD | 1.10 | (0.79–1.53) | 0.561 | 1.73 | (1.08–2.76) | 0.022 | 0.81 | (0.52–1.28) | 0.372 |
| **Women** | 1.05 | (0.92–1.19) | 0.471 | - | - | | - | - | |
| **Age** | 1.18 | (1.15–1.21) | <0.001 | 1.17 | (1.13–1.21) | <0.001 | 1.20 | (1.16–1.23) | <0.001 |
| **Born outside Sweden** | 0.79 | (0.65–0.96) | 0.018 | 0.64 | (0.47–0.88) | 0.005 | 0.93 | (0.73–1.20) | 0.600 |
| **Single, divorced, widowed** | 0.71 | (0.62–0.80) | <0.001 | 0.90 | (0.74–1.08) | 0.266 | 0.58 | (0.49–0.70) | <0.001 |
| **Education** | | | | | | | | | |
| Primary | 1.42 | (1.15–1.75) | 0.001 | 1.21 | (0.89–1.65) | 0.214 | 1.59 | (1.20–2.10) | 0.001 |
| Secondary | 1.43 | (1.23–1.66) | <0.001 | 1.37 | (1.11–1.69) | 0.003 | 1.47 | (1.19–1.82) | <0.001 |
| University | 1.00 | | | 1.00 | | | 1.00 | | |
| **Social occupational class** | | | | | | | | | |
| High non-manual | 1.00 | | | 1.00 | | | 1.00 | | |
| Intermediate non-manual | 1.03 | (0.88–1.20) | 0.740 | 0.90 | (0.72–1.13) | 0.365 | 1.21 | (0.97–1.51) | 0.087 |
| Low non-manual | 0.78 | (0.64–0.95) | 0.013 | 0.64 | (0.43–0.96) | 0.030 | 0.85 | (0.66–1.10) | 0.206 |
| Manual | 0.60 | (0.48–0.76) | <0.001 | 0.56 | (0.43–0.74) | <0.001 | 0.78 | (0.50–1.20) | 0.256 |
| **Financial strain** | 0.90 | (0.64–1.27) | 0.561 | 1.19 | (0.75–1.89) | 0.468 | 0.76 | (0.46–1.25) | 0.275 |
| **Employment conditions** | | | | | | | | | |
| Employed, great work freedom | 1.00 | | | 1.00 | | | 1.00 | | |
| Employed, limited work freedom | 1.43 | (1.20–1.71) | <0.001 | 1.17 | (0.84–1.64) | 0.350 | 1.57 | (1.27–1.95) | <0.001 |
| Self-employed | 1.23 | (1.02–1.48) | 0.027 | 1.22 | (0.98–1.52) | 0.082 | 1.25 | (0.90–1.73) | 0.188 |

SHR: Subdistribution hazard ratio, LLI; limiting longstanding illness, CMD: common mental disorder. All models adjust for all variables presented in the table.

hazard rate of early retirement. Adjusting for other socio-economic factors, men in the manual and low non-manual social occupational classes had a lower subdistribution hazard rate of early retirement compared to high non-manual social occupational class (SHR: 0.56, 0.64, respectively). In contrast, low education increased the subdistribution hazard rate of early retirement for women (SHR: 1.61 and 1.47, for primary and secondary education, respectively).

## Disability pension

Table 3 shows that LLI only (SHR: 11.43), and comorbid LLI+CMD (SHR: 14.25) was strongly associated with disability pension in the full sample. CMD only was also associated with disability pension, but was substantially lower compared to LLI only and comorbid LLI+CMD (SHR: 1.50, Table 3). This discrepancy is also evident from the CIFs presented in Fig 2. Post-hoc tests indicated that subdistribution hazard rate for LLI+CMD was not different from LLI only (SHR: 1.25, 95% CI: 0.99–1.57, p = 0.063; analyses not shown), but the subdistribution hazard rates for LLI only and LLI+CMD were substantially higher compared to CMD only (SHR for LLI only: 5.72, 95% CI: 3.82–8.56, p<0.001; SHR for LLI+CMD: 7.13, 95% CI: 4.60–11.05, p<0.001; analysis not shown). These associations did not differ by gender, but women were more likely to exit employment through disability pension. Education was not associated

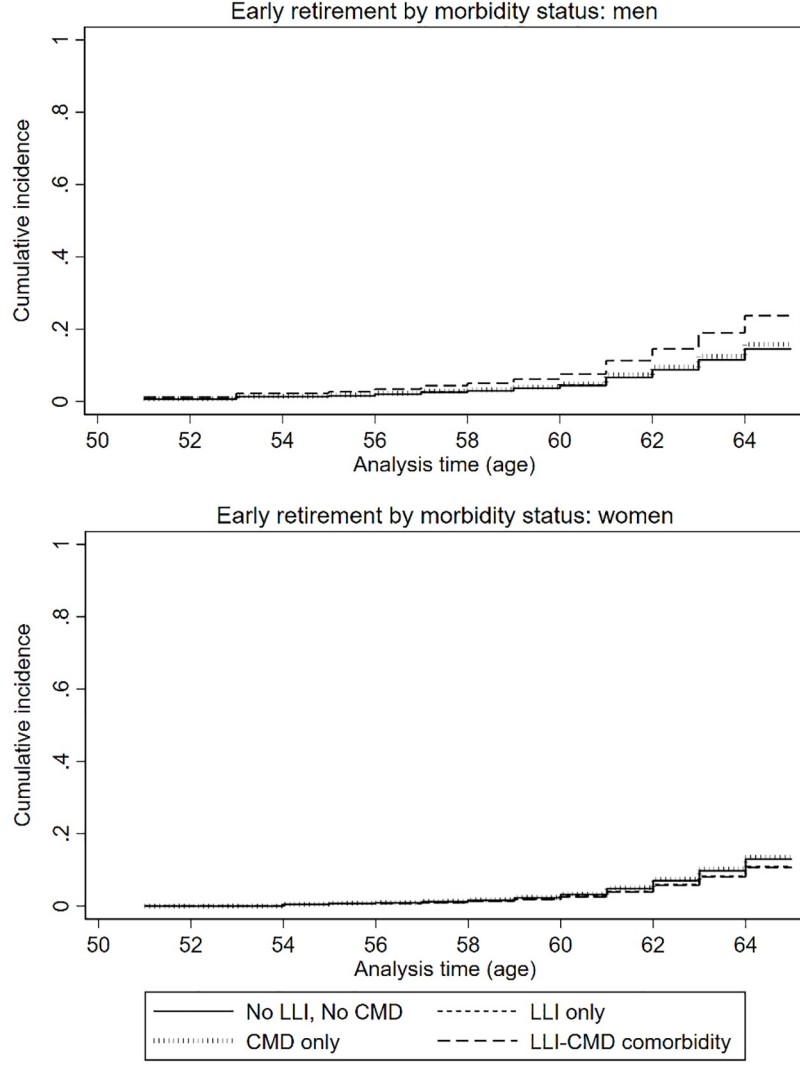

**Fig 1. The cumulative incidence functions (CIFs) of early retirement by morbidity status, where disability pension and unemployment are competing events.**

with disability pension for neither men nor women, but low social occupational class increased the subdistribution hazard rate of disability pension, specifically for men.

## Unemployment

LLI only and comorbid LLI+CMD were not associated with unemployment, but those with CMD only had an elevated subdistribution hazard rate of unemployment (SRH: 1.70, Table 4, Fig 3). These associations were similar in men and women. Older age was associated with a higher subdistribution hazard rate of unemployment. Migrants and those of divorced, single or widowed marital status, low social occupational class and with financial strain also had increased subdistribution hazard rates of unemployment. Unadjusted analyses (S1 Table) indicated that low education was strongly associated with unemployment, however social occupational class fully accounted for these associations in the adjusted model (stepwise adjustment not shown).

**Table 3. Adjusted competing risks analyses on the influence of health, demographic, and socio-economic factors and work conditions at baseline among employed persons on the likelihood of disability pension in the full sample, and stratified by gender.**

| | Disability pension: Full sample (n = 578/n = 9811) | | | Disability pension: Men (n = 177/4600) | | | Disability pension: Women (n = 401/5211) | | |
|---|---|---|---|---|---|---|---|---|---|
| | SHR | (95% CI) | *p* | SHR | (95% CI) | *p* | SHR | (95% CI) | *p* |
| **Morbidity** | | | | | | | | | |
| No LLI, No CMD | 1.00 | | | 1.00 | | | 1.00 | | |
| LLI only | 11.43 | (9.40–13.89) | <0.001 | 11.05 | (7.79–15.68) | <0.001 | 11.60 | (9.15–14.70) | <0.001 |
| CMD only | 2.00 | (1.31–3.05) | 0.001 | 2.32 | (0.99–5.42) | 0.053 | 1.89 | (1.16–3.08) | 0.011 |
| LLI+CMD | 14.25 | (10.91–18.61) | <0.001 | 15.11 | (8.76–26.08) | <0.001 | 14.30 | (10.53–19.41) | <0.001 |
| **Women** | 1.92 | (1.57–2.35) | <0.001 | - | | | - | | |
| **Age** | 1.65 | (1.56–1.75) | <0.001 | 1.58 | (1.42–1.75) | <0.001 | 1.69 | (1.58–1.81) | <0.001 |
| **Born outside Sweden** | 1.02 | (0.82–1.26) | 0.889 | 0.82 | (0.54–1.23) | 0.337 | 1.10 | (0.85–1.42) | 0.465 |
| **Single, divorced, widowed** | 1.01 | (0.86–1.20) | 0.884 | 0.91 | (0.66–1.25) | 0.560 | 1.06 | (0.87–1.42) | 0.577 |
| **Education** | | | | | | | | | |
| Primary | 1.03 | (0.78–1.34) | 0.856 | 1.11 | (0.68–1.79) | 0.683 | 1.01 | (0.87–1.29) | 0.948 |
| Secondary | 0.86 | (0.69–1.07) | 0.180 | 1.03 | (0.69–1.53) | 0.888 | 0.81 | (0.62–1.06) | 0.126 |
| Higher education | 1.00 | | | 1.00 | | | 1.00 | | |
| **Social occupational class** | | | | | | | | | |
| High non-manual | 1.00 | | | 1.00 | | | 1.00 | | |
| Intermediate non-manual | 1.30 | (1.02–1.64) | 0.031 | 1.64 | (1.03–2.61) | 0.036 | 1.17 | (0.89–1.55) | 0.253 |
| Low non-manual | 1.39 | (1.06–1.83) | 0.016 | 2.09 | (1.19–3.69) | 0.011 | 1.28 | (0.93–1.74) | 0.127 |
| Manual | 1.67 | (1.24–2.25) | 0.001 | 1.88 | (1.18–2.99) | 0.008 | 1.55 | (0.99–2.42) | 0.057 |
| **Financial strain** | 1.19 | (0.89–1.59) | 0.252 | 1.54 | (0.90–2.64) | 0.112 | 1.08 | (0.76–1.53) | 0.662 |
| **Employment conditions** | | | | | | | | | |
| Employed, great work freedom | 1.00 | | | 1.00 | | | 1.00 | | |
| Employed, limited work freedom | 1.48 | (1.20–1.83) | <0.001 | 1.98 | (1.32–2.95) | 0.001 | 1.35 | (1.06–1.72) | 0.016 |
| Self-employed | 0.96 | (0.72–1.29) | 0.794 | 1.47 | (1.00–2.14) | 0.048 | 0.57 | (0.34–0.97) | 0.037 |

SHR: Subdistribution hazard ratio, LLI; limiting longstanding illness, CMD: common mental disorder. All models adjust for all variables presented in the table.

## Discussion

This study found strong health effects for specific morbidity categories and different employment exits. These findings are consistent with the literature indicating that the association between poor health and employment exit depends on the pathway [3,8,29]. Our findings suggest that comorbid LLI+CMD was important for early retirement specifically in men, that LLI—with or without CMD—was the major determinant of disability pension in both men and women, while CMD without LLI was specifically associated with unemployment. These findings are supported by a recent study which examined competing employment exits among older Dutch workers and observed similar results—strong effects of comorbidity on disability pension, no health effects on early retirement, and a specific association between mental illness and unemployment [8]. Our study extends this research in providing novel insights into how the comorbidity of LLI's and CMD's affects competing employment exits in older working-age men and women, noting for example that the combination of LLI+CMD is a relevant determinant of early retirement for men, but not women.

### Findings in context with the literature

The absence of an association between any of the morbidity categories with early retirement in the full sample is consistent with past research observing that poor self-rated health, mental

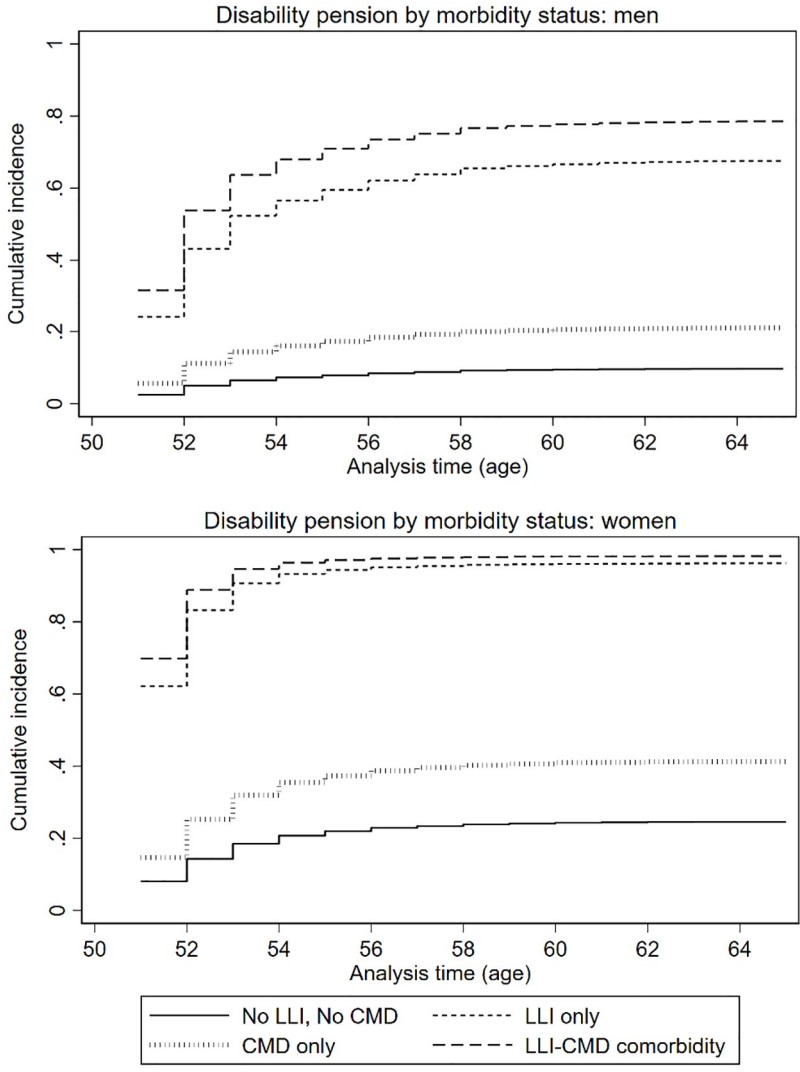

**Fig 2. The cumulative incidence functions (CIFs) of disability pension by morbidity status, where early retirement and unemployment are competing events.**

illness and chronic illness is less strongly associated with early retirement as opposed to disability pension and unemployment [3,8,30]. It is also in line with a study of Dutch older workers which failed to observe amplified effects of comorbid depression on early retirement among workers with chronic illness, compared to workers without chronic illness [4]. Contrary to research from other countries, women were no more likely to retire earlier than men [2]. However, the drivers for early retirement seemed to vary by gender; comorbid LLI+CMD was specifically associated with early retirement for men, but none of the morbidity categories were associated with early retirement for women. This is consistent with studies which have observed that poor health is a more common reason for involuntary employment exits for men than women [29,31]. This suggests that women who retire early do so for reasons other than being pushed into early retirement for health reasons. For example, social and familial pull factors may be more important for women's early retirement decisions, including their partners' employment status and caring demands [19].

**Table 4. Adjusted competing risks analyses on the influence of health, demographic, and socio-economic factors and work conditions at baseline among employed persons on the likelihood of unemployment in the full sample, and stratified by gender.**

| | Unemployment: full sample (n = 806/ n = 9811) | | | Unemployment: men (n = 377/4600) | | | Unemployment: women (n = 429/5211) | | |
|---|---|---|---|---|---|---|---|---|---|
| | SHR | (95% CI) | p | SHR | (95% CI) | p | SHR | (95% CI) | p |
| **Morbidity** | | | | | | | | | |
| No LLI, No CMD | 1.00 | | | 1.00 | | | 1.00 | | |
| LLI only | 1.00 | (0.83–1.22) | 0.981 | 1.17 | (0.90–1.53) | 0.247 | 0.87 | (0.65–1.15) | 0.327 |
| CMD only | 1.70 | (1.36–2.15) | <0.001 | 1.81 | (1.24–2.67) | 0.002 | 1.64 | (1.23–2.19) | 0.001 |
| LLI+CMD | 0.96 | (0.66–1.41) | 0.848 | 1.25 | (0.68–2.30) | 0.467 | 0.83 | (0.51–1.35) | 0.457 |
| **Women** | 0.90 | (0.76–1.05) | 0.185 | - | | | - | | |
| **Age** | 1.28 | (1.23–1.33) | <0.001 | 1.27 | (1.20–1.34) | <0.001 | 1.29 | (1.22–1.35) | <0.001 |
| **Born outside Sweden** | 1.29 | (1.08–1.55) | 0.005 | 1.30 | (1.00–1.70) | 0.054 | 1.26 | (0.99–1.60) | 0.058 |
| **Single, divorced, widowed** | 1.20 | (1.04–1.39) | 0.005 | 1.27 | (1.03–1.56) | 0.025 | 1.15 | (0.95–1.40) | 0.151 |
| **Education** | | | | | | | | | |
| Primary | 1.07 | (0.83–1.37) | 0.603 | 1.06 | (0.75–1.50) | 0.740 | 1.08 | (0.75–1.54) | 0.681 |
| Secondary | 1.12 | (0.93–1.36) | 0.242 | 1.22 | (0.93–1.59) | 0.151 | 1.06 | (0.79–1.40) | 0.709 |
| Higher education | 1.00 | | | 1.00 | | | 1.00 | | |
| **Social occupational class** | | | | | | | | | |
| High non-manual | 1.00 | | | 1.00 | | | 1.00 | | |
| Intermediate non-manual | 1.31 | (1.06–1.62) | 0.011 | 1.40 | (1.04–1.89) | 0.028 | 1.23 | (0.92–1.64) | 0.161 |
| Low non-manual | 1.86 | (1.45–2.38) | <0.001 | 2.29 | (1.58–3.32) | <0.001 | 1.75 | (1.25–2.45) | 0.001 |
| Manual | 1.77 | (1.37–2.29) | <0.001 | 1.68 | (1.21–2.33) | 0.002 | 2.29 | (1.46–3.58) | <0.001 |
| **Financial strain** | 1.30 | (1.01–1.69) | 0.042 | 1.36 | (0.92–2.00) | 0.126 | 1.26 | (0.90–1.79) | 0.180 |
| **Employment conditions** | | | | | | | | | |
| Employed, great work freedom | 1.00 | | | 1.00 | | | 1.00 | | |
| Employed, limited work freedom | 1.27 | (1.05–1.54) | 0.013 | 1.31 | (0.97–1.76) | 0.083 | 1.25 | (0.98–1.69) | 0.079 |
| Self-employed | 0.83 | (0.64–1.07) | 0.149 | 0.71 | (0.51–0.99) | 0.041 | 1.16 | (0.78–1.73) | 0.457 |

SHR: Subdistribution hazard ratio, LLI; limiting longstanding illness, CMD: common mental disorder. All models adjust for all variables presented in the table.

The finding that none of the morbidity categories were associated with early retirement in women may also partially be explained by women in poor health being more likely obtain disability pension. This observation supports past research in Sweden [32] and national statistics indicating that more women were granted disability pension for this time period than men [33]. Whilst women were at greater risk of disability pension, the health determinants for women and men were similar. Consistent with previous research, strong associations between LLI only and comorbid LLI+CMD were observed with disability pension, [34] however, there was no amplified effect of comorbidity on disability pension. This contradicts research using Swedish healthcare register data, where comorbid heart disease and diagnosed depression had an amplified effect on disability pension [10]. This inconsistency may be explained by health-care records capturing and more severe diagnosed conditions, in contrast to the symptom screen used in our study.

We found that men and women with CMD only were particularly vulnerable to unemployment, while no such associations were observed for LLI only and comorbid LLI+CMD. Poor mental health has previously been found to be associated with employment exits in this age group [4,6,8]. By examining the specific overlap between LLI and CMD, our study indicated that CMD in the absence of an LLI may represent a barrier to obtaining disability pension,

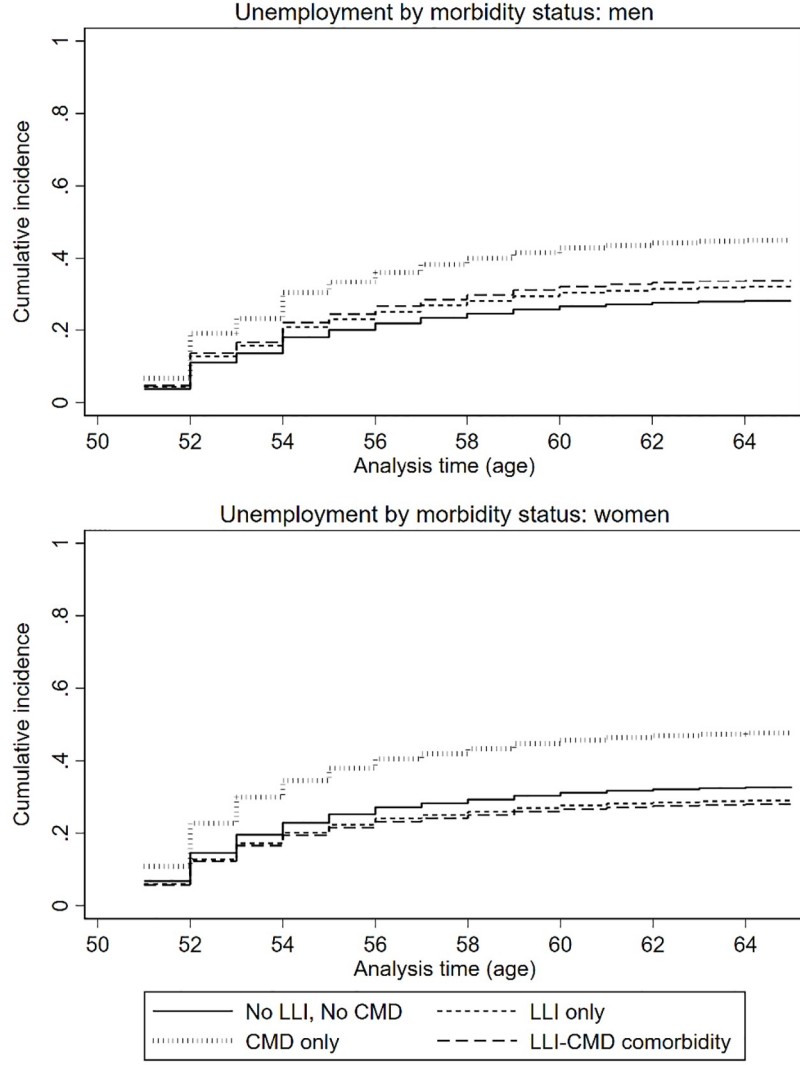

**Fig 3. The cumulative incidence functions (CIFs) of unemployment by morbidity status, where early retirement and disability pension are competing events.**

which may make unemployment a more likely employment exit route for those with occupational impairments brought on by a CMD.

The results highlighted other important determinants of employment exits, independent of health, which were also specific to employment exit routes. Being married increased the likelihood of early retirement for women, consistent with studies observing that having a spouse represents an important pull factor into retirement [6,35]. Men in lower social occupational classes had a lower subdistribution hazard rate of retiring early. In part, this may be due to that those with higher social occupational class have better financial assets and accumulated pension which may enable them to retire early, while financial barriers may prevent men in lower social occupational classes from doing so. Such financial factors may not apply to the same extent to women, who may be more likely to receive support from husbands with higher pensions. Social gradients were observed such that workers with lower education and lower social occupational class were more likely to exit through disability pension and unemployment [36,37]. Migrants were less likely to retire early, but at greater risk of unemployment; a finding

consistent with Canadian research which points to a potentially important inequality in employment exits by migration status [9]. Furthermore, limited freedom at work was a determinant of all employment exit routes, consistent with evidence pointing to the importance of good psychosocial working conditions for older workers to stay in employment [4,30,38].

## Policy context and implications

Research studying employment, early retirement and disability pension in Sweden over this time period suggests that many with chronic illness were forced out of employment through alternative exit routes, due to eligibility restrictions of disability pension [39,40]. Taken together with the finding that women were more likely to obtain disability pension than men, our findings may suggest that comorbidity acted as a push factor of early retirement for men, forcing them into early retirement due to failures to qualify for disability pension. Similarly, since CMD only was not as strongly associated with disability pension as LLI only and LLI +CMD, but specifically associated with unemployment, it may suggest that those with CMD, in the absence of LLI, are less likely to qualify for disability pension and instead become unemployed [40,41]. This would mean that further restrictions to disability pension are unlikely to facilitate extended working lives; rather, those too unwell to work would exit through a different route instead [40]. To extend working lives, policies may instead focus on supporting older workers to stay in employment through tailored workplace adaptations to address the diversity of older workers' needs. This may also reduce health inequalities in older ages, given that good employment has health benefits for older workers [42,43]. Qualitative research of Swedish white-collar workers found that employer support, task-shifting and early adaptation, could facilitate older workers with chronic diseases to continue working [43].

We also identified that workers with CMD without LLI, low education, and migrant background may be particularly vulnerable to employment exits. Since financial wealth (e.g. future pension) often increases with the length of employment, early employment exits for these groups may contribute to socio-economic inequalities in older ages, and therefore ought to be considered in future policy. The fact that specific morbidity combinations were associated with particular exit routes among workers aged 50–62, suggests that policies aimed at extending working lives need to be tailored to targeted employment exit routes, and that different approaches are needed to support men and women to stay in employment. However, given that the association between health and specific employment exit routes are influenced by policy context [3], the generalisability of the results is likely to be limited to countries with similar pension and disability policies as Sweden.

## Limitations

Our sample is likely to consist of workers who are relatively healthy, wealthy and high functioning, due to health-selection effects such that healthy older workers are more likely to be in employment over the age of 50, and due to selective non-participation in the initial survey sample [21]. The effect of comorbid LLI+CMD on employment exits may therefore have been underestimated due to selection bias. Studying those aged 50–62 in employment nevertheless makes the findings relevant for the purposes of informing extending working lives policy. Whilst our comorbidity measure captured important dimensions of illness burden including chronicity, disability and psychiatric symptoms; counts of LLI's or specific diagnoses combinations were not captured. Since dose-response associations between the accumulation of chronic illnesses and employment exits have been observed [6,14–16], a comorbidity measure which captured the number of LLI's may have identified stronger associations with specific employment exit routes. Furthermore, the measure of LLI did not exclude mental disorders,

which could have led to some misclassification in the morbidity categories. Moreover, whilst the GHQ-12 is a well-validated screening tool for CMDs, it does not accurately indicate diagnosis. Nevertheless, the benefit of using a screening tool in a community population sample is that it is likely to capture those with undiagnosed CMDs who are not in contact with services.

## Conclusions

Maintaining good health in older workers is one of the most important issues to address in policies aiming to extend working lives. Our study indicated that health determinants of employment exits were different for men and women and specific to particular exit routes. Men with comorbidity were more likely to exit employment through disability pension and early retirement, whilst women were more likely to obtain disability pension only. We also found that men and women with CMD only may be particularly vulnerable to unemployment. Initiatives to extend working lives should therefore be tailored to targeted exit routes, and consider the varied health needs of older workers.

## Supporting information

**S1 Fig. Process of deriving the analytical sample.**
(DOCX)

**S1 Table. Unadjusted competing risks analyses on the influence of health, demographic, and socio-economic factors and work conditions at baseline among employed persons on the likelihood of employment exit (N = 10,416).**
(DOCX)

## Acknowledgments

We would like to thank the Tackling Health Inequalities and Extending Working Lives (THRIVE) team and the members of the Equity and Health Policy Research Group for their constructive comments in the making of this manuscript.

## Author Contributions

**Conceptualization:** Lisa Harber-Aschan, Wen-Hao Chen, Bo Burström.

**Formal analysis:** Lisa Harber-Aschan, Wen-Hao Chen.

**Funding acquisition:** Finn Diderichsen, Ben Barr, Bo Burström.

**Methodology:** Lisa Harber-Aschan.

**Project administration:** Ashley McAllister.

**Supervision:** Bo Burström.

**Visualization:** Lisa Harber-Aschan.

**Writing – original draft:** Lisa Harber-Aschan.

**Writing – review & editing:** Lisa Harber-Aschan, Wen-Hao Chen, Ashley McAllister, Natasja Koitzsch Jensen, Karsten Thielen, Ingelise Andersen, Finn Diderichsen, Ben Barr, Bo Burström.

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
