## [Decision Letter · Decision Letter 0]

25 Nov 2019

PONE-D-19-28997

The impact of longstanding illness and common mental disorder on competing employment exits routes in older working age: a longitudinal data-linkage study in Sweden

PLOS ONE

Dear Dr. Lisa Harber-Aschan,

Thank you for submitting your manuscript to PLOS ONE. After careful consideration, we feel that it has merit but does not fully meet PLOS ONE’s publication criteria as it currently stands. Therefore, we invite you to submit a revised version of the manuscript that addresses the points raised during the review process.

ACADEMIC EDITOR: The reviewers have raised a number of points which we believe major modifications are necessary to improve the manuscript, taking into account the reviewers' remarks.  Please consider and address each of the comments raised by the reviewers before resubmitting the manuscript. This letter should not be construed as implying acceptance, as a revised version will be subject to re-review.

We would appreciate receiving your revised manuscript by Jan 09 2020 11:59PM. To enhance the reproducibility of your results, we recommend that if applicable you deposit your laboratory protocols in protocols.io, where a protocol can be assigned its own identifier (DOI) such that it can be cited independently in the future. For instructions see: http://journals.plos.org/plosone/s/submission-guidelines#loc-laboratory-protocols

We look forward to receiving your revised manuscript.

Kind regards,

Wisit Cheungpasitporn, MD, FACP

Academic Editor

PLOS ONE

Journal Requirements:

1. Thank you for including your ethics statement:

"The study was approved by the Regional Ethical Review Board in Stockholm (2016/1353-31/5). The study was a secondary data analysis which analysed data anonymously.".

i) Please amend your current ethics statement to include the full name of the ethics committee/institutional review board(s) that approved your specific study.

ii) Once you have amended this/these statement(s) in the Methods section of the manuscript, please add the same text to the “Ethics Statement” field of the submission form (via “Edit Submission”).

2. Please be wary of making any causal inferences from this study, due to its nature and design. For example, you state "Comorbidity pushed men into disability pension and early retirement", which cannot be supported by this study design.

Reviewers' comments:

Reviewer's Responses to Questions

**Comments to the Author**

1. Is the manuscript technically sound, and do the data support the conclusions?

Reviewer #1: Yes

Reviewer #2: Yes

Reviewer #3: Yes

2. Has the statistical analysis been performed appropriately and rigorously? 

Reviewer #1: Yes

Reviewer #2: I Don't Know

Reviewer #3: No

3. Have the authors made all data underlying the findings in their manuscript fully available?

Reviewer #1: Yes

Reviewer #2: No

Reviewer #3: No

4. Is the manuscript presented in an intelligible fashion and written in standard English?

Reviewer #1: Yes

Reviewer #2: Yes

Reviewer #3: Yes

5. Review Comments to the Author

Reviewer #1: In this study, authors examined the association between comorbidity types and employment exit routes. I feel interesting the findings that limiting longstanding illness, common mental disorder, and the combination differently influenced employment exit routes. I think that the sample was sufficient, and the analysis was appropriate. However, I think that authors need to clarify the definition of common mental disorder. I listed the comments to the following.

Major points

“Mental illness is also recognised as an important…” (L51)

I think that the examples of mental illness (e.g., depression, schizophrenia, etc.) help readers.

“CMD was captured using the 12-item General Health Questionnaire (GHQ-12), a screen assessing symptoms of psychological distress” (L95-96)

GHQ-12 is a screening tool, which can not elucidate accurate diagnosis. Thus, the points should be added to the limitation section. In addition, CMD is a broad concept, whereas GHQ-12 can not capture all of CMD. For example, I think that GHQ-12 was not related to schizophrenia and dementia. I think that authors should describe the definition of CMD in this study.

Minor points

“and was recently was found to be specifically associated with unemployment in” (L52)

Exclude the second “was” in the sentence.

Reviewer #2: The subject and the matter are an ongoing issue in modern society which is directly affecting the mental health state and physical health of the relevant population and also their families and colleagues.

This subject is covering not only mental health but also the socio-economic aspects of employment and also occupational health. I as a psychiatrist would like to see more work form the authors with possibility of same studies in different countries.

Reviewer #3: The authors have identified good problem. I suggest to include the following points in your statistical analysis:

1) Graphical representation of the results.

2) regression analysis should also be included.

6. PLOS authors have the option to publish the peer review history of their article (what does this mean?). If published, this will include your full peer review and any attached files.

Reviewer #1: No

Reviewer #2: Yes: Dr Lily Abedipour MD

Reviewer #3: No

---

## [Author Response · Author response to Decision Letter 0]

9 Jan 2020

Response to the editor’s comments: 

1. Thank you for including your ethics statement: "The study was approved by the Regional Ethical Review Board in Stockholm (2016/1353-31/5). The study was a secondary data analysis which analysed data anonymously.".

i) Please amend your current ethics statement to include the full name of the ethics committee/institutional review board(s) that approved your specific study.

ii) Once you have amended this/these statement(s) in the Methods section of the manuscript, please add the same text to the “Ethics Statement” field of the submission form (via “Edit Submission”).

The ethical review board name stated in the manuscript name is the full name of the ethical review board translated into English. In Swedish the name is: Regionala Etikprövningsnämnden (EPN) i Stockholm. At the time of the study, the organisation for ethical review of research involving humans consisted of a total of seven boards in Sweden. Ethical reviews took place at seven regional ethical review boards, and we applied to the board in Stockholm. In 2019 the regional ethical review boards were replaced with one central ethical review institution called the “Swedish Ethical Review Authority” or “Etikprövningsmyndigheten” in Swedish. 

To clarify, we have edited the ethics statement to include the full name Swedish (Regionala Etikprövningsnämnden (EPN) i Stockholm) in the online submission form, and added the ethics statement to the methods section of the manuscript (p. 5, line 93-95). 

2. Please be wary of making any causal inferences from this study, due to its nature and design. For example, you state "Comorbidity pushed men into disability pension and early retirement", which cannot be supported by this study design.

We appreciate that this statement implied causality and have changed it to merely comment on the associations observed: “Men with comorbidity were more likely to exit employment through disability pension and early retirement” (p 18, line 272-263).

Accordingly, we also changed the statement: “pushed into unemployment” to “become unemployed” (p. 16, line 237). 

Response to reviewers: 

Reviewer #1: 

In this study, authors examined the association between comorbidity types and employment exit routes. I feel interesting the findings that limiting longstanding illness, common mental disorder, and the combination differently influenced employment exit routes. I think that the sample was sufficient, and the analysis was appropriate. However, I think that authors need to clarify the definition of common mental disorder. I listed the comments to the following.

Major points

“Mental illness is also recognised as an important…” (L51)

I think that the examples of mental illness (e.g., depression, schizophrenia, etc.) help readers.

We agree with the reviewer that “mental illness” is broad term. For the purposes of this particular study we have focused on depressive and anxiety disorders that are typically referred to as common mental disorders (CMDs), many of which are prevalent in the community samples such as ours (Goldberg & Huxley, 1992; McManus, et al., 2009). We have therefore stated this early on in the discussion, changing the term “mental illness” to “common mental disorders” (p 4, line 51-54). 

“CMD was captured using the 12-item General Health Questionnaire (GHQ-12), a screen assessing symptoms of psychological distress” (L95-96)

GHQ-12 is a screening tool, which can not elucidate accurate diagnosis. Thus, the points should be added to the limitation section. In addition, CMD is a broad concept, whereas GHQ-12 can not capture all of CMD. For example, I think that GHQ-12 was not related to schizophrenia and dementia. I think that authors should describe the definition of CMD in this study.

It was indeed unclear that our focus were depression and anxiety disorders (which we refer to as CMDs). As stated above, CMDs have now been defined early in the introduction (p 4, line 51-54). The GHQ is a well validated tool which readily captures depression and anxiety diagsnoses as validated by outpatient records for this specific sample (Lundin et al., 2017, cited in the manuscript on p. 6, line 99-100, reference no. 23). We have also added a sentence to the strengths and limitations section, acknowledging its limitations, but also arguing that there may be some benefits to using screening tools as this may capture undiagnosed cases of CMDs unknown to services (p. 19, line 265-268).

Minor points

“and was recently was found to be specifically associated with unemployment in” (L52)

Exclude the second “was” in the sentence.

This typo has now been edited (p 4, line 53). 

Reviewer #2: 

The subject and the matter are an ongoing issue in modern society which is directly affecting the mental health state and physical health of the relevant population and also their families and colleagues.

This subject is covering not only mental health but also the socio-economic aspects of employment and also occupational health. I as a psychiatrist would like to see more work form the authors with possibility of same studies in different countries.

We thank the reviewer for this comment, and we can confirm that we are working on cross-national comparative studies. 

Reviewer #3: 

The authors have identified good problem. I suggest to include the following points in your statistical analysis:

1) Graphical representation of the results.

We would like to thank the reviewer for this suggestion. We have now added graphs showing the cumulative incidence function by morbidity status, for men and women (Figures 1, 

2 and 3), which we believe make an important development to the paper. These graphs do not only complement the tables to visualise the results, but also clearly demonstrate the how the employment exit types differ from one another. 

We added a couple of sentences to the analysis section describing cumulative incidence functions (p. 7, line 139-141), and make reference to each of the figures in turn for the respective type of employment exit discussed in the results, along with figure appropriate captions (p. 10, line 139, 146-147; p. 12; line 151-152, 160-161; p. 14, line 164, 171-172). 

2) regression analysis should also be included.

The current models that we present are indeed types of regression models. Fine and Gray subdistribution hazard models are similar to cox regression models but allow for considering competing events. 

References 

Goldberg D, Huxley P. Common Mental Disorders: A biosocial model. London: Travistock/Routledge 1992.

Lundin A, Forsell Y, Dalman C. Mental health service use, depression, panic disorder and life events among Swedish young adults in 2000 and 2010: a repeated cross-sectional population study in Stockholm County, Sweden. Epidemiol Psychiatr Sci. 2017; 1–9. doi:10.1017/S2045796017000099

McManus S, Melzer H, Brugha T, Bebbington P, Jenkins R. Adult Psychiatric Morbidity Survey in England, 2007: Results of a Household Survey. London: National Centre for Social Research 2009.

---

## [Decision Letter · Decision Letter 1]

3 Feb 2020

The impact of longstanding illness and common mental disorder on competing employment exits routes in older working age: a longitudinal data-linkage study in Sweden

PONE-D-19-28997R1

Dear Dr. Lisa Harber-Aschan,

We are pleased to inform you that your manuscript has been judged scientifically suitable for publication and will be formally accepted for publication once it complies with all outstanding technical requirements.

With kind regards,

Wisit Cheungpasitporn, MD, FACP

Academic Editor

PLOS ONE

Additional Editor Comments:

I reviewed the revised manuscript and the response to reviewers' comments. Revised Manuscript is well written. All comments have been addressed and thus accepted for publication.

Reviewers' comments:

Reviewer's Responses to Questions

**Comments to the Author**

1. If the authors have adequately addressed your comments raised in a previous round of review and you feel that this manuscript is now acceptable for publication, you may indicate that here to bypass the “Comments to the Author” section, enter your conflict of interest statement in the “Confidential to Editor” section, and submit your "Accept" recommendation.

Reviewer #2: All comments have been addressed

Reviewer #4: All comments have been addressed

2. Is the manuscript technically sound, and do the data support the conclusions?

Reviewer #2: Yes

Reviewer #4: Yes

3. Has the statistical analysis been performed appropriately and rigorously? 

Reviewer #2: I Don't Know

Reviewer #4: Yes

4. Have the authors made all data underlying the findings in their manuscript fully available?

Reviewer #2: Yes

Reviewer #4: Yes

5. Is the manuscript presented in an intelligible fashion and written in standard English?

Reviewer #2: Yes

Reviewer #4: Yes

6. Review Comments to the Author

Reviewer #2: It seems that the authors have addressed the outlined issues and have also added further explanations.

Reviewer #4: I have no competing interests. I thank the author(s) for addressing questions and concerns. Given the changes made and the major concerns being addressed, I have no further reservations regarding publication of the revised version of the manuscript.

7. PLOS authors have the option to publish the peer review history of their article (what does this mean?). If published, this will include your full peer review and any attached files.

Reviewer #2: Yes: Dr Lily Abedipour MD

Reviewer #4: No

---

## [Editor Report · Acceptance letter]

11 Feb 2020

PONE-D-19-28997R1 

The impact of longstanding illness and common mental disorder on competing employment exits routes in older working age: a longitudinal data-linkage study in Sweden 

Dear Dr. Harber-Aschan:

I am pleased to inform you that your manuscript has been deemed suitable for publication in PLOS ONE. Congratulations! Your manuscript is now with our production department. 

With kind regards,

on behalf of

Dr. Wisit Cheungpasitporn 

Academic Editor

PLOS ONE